# Maximum *a posteriori* natural scene reconstruction from retinal ganglion cells with deep denoiser priors

**Eric G. Wu**
Stanford University
wueric@stanford.edu

**Nora Brackbill**
Stanford University

**Alexander Sher & Alan M. Litke**
University of California, Santa Cruz

**Eero P. Simoncelli**
Flatiron Institute, Simons Foundation,
and New York University
eero.simoncelli@nyu.edu

**E.J. Chichilnisky**
Stanford University
ej@stanford.edu

## Abstract

Visual information arriving at the retina is transmitted to the brain by signals in the optic nerve, and the brain must rely solely on these signals to make inferences about the visual world. Previous work has probed the content of these signals by directly reconstructing images from retinal activity using linear regression or nonlinear regression with neural networks. Maximum *a posteriori* (MAP) reconstruction using retinal encoding models and separately-trained natural image priors offers a more general and principled approach. We develop a novel method for approximate MAP reconstruction that combines a generalized linear model for retinal responses to light, including their dependence on spike history and spikes of neighboring cells, with the image prior implicitly embedded in a deep convolutional neural network trained for image denoising. We use this method to reconstruct natural images from *ex vivo* simultaneously-recorded spikes of hundreds of retinal ganglion cells uniformly sampling a region of the retina. The method produces reconstructions that match or exceed the state-of-the-art in perceptual similarity and exhibit additional fine detail, while using substantially fewer model parameters than previous approaches. The use of more rudimentary encoding models (a linear-nonlinear-Poisson cascade) or image priors (a $1/f$ spectral model) significantly reduces reconstruction performance, indicating the essential role of both components in achieving high-quality reconstructed images from the retinal signal.

## 1 Introduction

A torrent of visual information arrives at each of our eyes where it is transduced by a large population of photoreceptors and transformed by retinal circuitry. The resulting signals are transmitted to the brain in the spikes of a relatively small number of retinal ganglion cells (RGCs), suggesting that a large fraction of this visual information is lost. Elucidating the nature of these encoded signals, and the inference processes the brain could use to interpret them, is fundamental to understanding biological vision. Image reconstruction provides a method of visualizing the content encoded by RGC signals, evaluating it using standard image quality metrics, and reasoning about how the brain might use it [1, 2, 3, 4]. The fidelity and quality of reconstructed images also provides a useful objective function for optimizing the design of electrical stimulation patterns delivered by devices implanted to restore vision to people blinded by retinal degeneration [5, 6].

36th Conference on Neural Information Processing Systems (NeurIPS 2022).

The simplest and most well-studied image reconstruction method is linear regression [1, 3, 7]. Optimal reconstruction kernels are learned for each RGC using least-squares regression of recorded responses to many visual images, and the reconstruction of a new incident image is computed with the sum of the filters weighted by the measured response of each cell. The quality of linearly reconstructed images can be enhanced by applying an autoencoder neural network to impose natural image priors [8], or by using deep neural networks to non-linearly recover additional high spatial frequency image components [4]. Neural networks can also be directly trained (supervised) for reconstruction from raw data, but this is data-intensive, and has limited their use to simulated data, or low-dimensional stimuli and small numbers of cells [9]. These direct approaches leave substantial room for improvement and interpretation. A Bayesian formulation, in which encoding model and image priors are made explicit and are separately designed and optimized, offers a more flexible and interpretable solution.

Here, we present a method for approximate maximum *a posteriori* (MAP) image reconstruction from recorded RGC spikes, combining a retinal encoding model that accurately captures retinal responses [10] with state-of-the-art image priors that are implicitly embedded in deep denoising networks [11, 12, 13, 14, 15, 16]. Separating the contributions of image prior and RGC spiking response likelihood allows independent training of both elements, and offers flexible exploration and interpretation. Specifically, (1) any pre-trained or closed-form natural image prior can be used, and the effects of different priors can be compared; and (2) any model of RGC encoding that provides an explicit likelihood can be used, allowing examination of the relative importance of different model components in representing the visual signal, such as spike train history, cell-to-cell correlations and output nonlinearities. We apply our method to reconstruct flashed natural images from responses of several hundred macaque RGCs of identified types recorded with large-scale electrode arrays. We compare these results to published state-of-the-art linear and neural network regression methods. The new method matches or outperforms previous methods, producing sharper, more naturalistic reconstructions, and similar or greater perceptual similarity to ground truth. However, our method also produces some reconstructions with distinctive hallucinated image structure, as would be expected when RGC signals are noisy and image priors dominate the reconstruction process. Finally, comparisons to reconstructions using more conventional encoding models and image priors reveal that both aspects of the approach are important for the most accurate reconstructions. Source code, fitted models, and a subset of the test dataset used to produce example reconstructions are available at https://github.com/wueric/RGC_MAP_reconstruction.

## 2 Retinal data and stimuli

Extracellular recordings from RGCs in the peripheral macaque retina were performed *ex vivo* using a 512-electrode system [17] (see [3] for details). Retinas were obtained from terminally anesthetized macaque monkeys used by other laboratories, in accordance with Institutional Animal Care and Use Committee requirements. Spikes from individual RGCs were identified with the YASS [18] spike-sorter. A 30-minute spatio-temporal white noise stimulus [19] was used to compute spatio-temporal receptive fields, and to identify cells of distinct types. Analysis focused on the four major RGC types of the primate retina (ON parasol, OFF parasol, ON midget, and OFF midget) [20, 21], totaling roughly 700 cells per recording. The receptive fields [22] of all four cell types formed regular mosaics, with each providing nearly-complete coverage of a region of visual space (Figure 1B).

Natural images were presented to the retina as described previously [3]. Images from the ImageNet database [23] were converted to grayscale and cropped to 256x160. Each pixel measured approximately $11 \times 11 \, \mu$m at the retina. Each stimulus image was displayed statically for 100 ms, followed by a 400 ms uniform gray display, allowing each image presentation to be treated as an independent trial (Figure 1A). Recordings were gathered from retinas of two animals, in response to 19,000 and 10,000 images, respectively. Details are summarized in Tables 4 and 5 in the Appendix.

## 3 MAP image reconstruction from RGC spikes

The MAP estimate of the stimulus image $\mathbf{x}$ (a vector of pixel luminances) from observed RGC spike trains $\mathbf{s}$ corresponds to the minimum of the negative log of the posterior distribution, $-\log p(\mathbf{x} \mid \mathbf{s})$, which can be expressed using Bayes' Rule as:

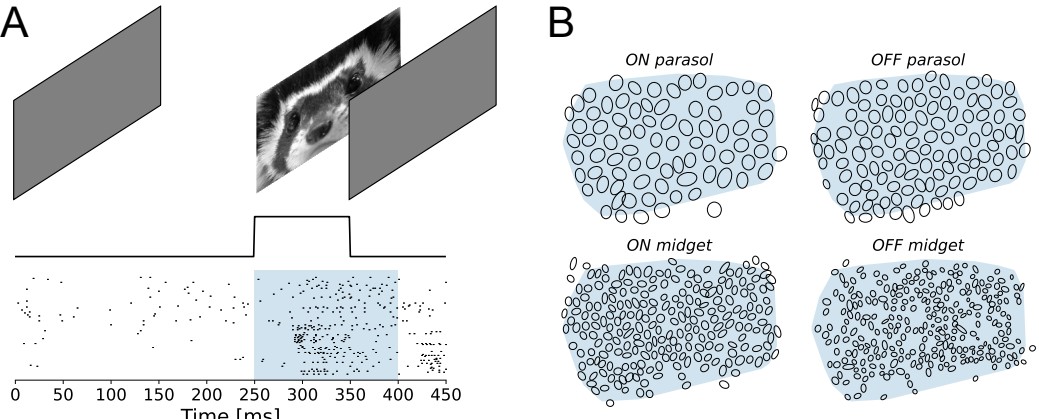

Figure 1: (A) Trial structure for a single stimulus image presentation. Static stimulus images are displayed for 100 ms, separated by 400 ms uniform gray display. Spikes occurring within 150 ms of the onset of the stimulus image, corresponding to the shaded blue region, were used to compute encoding negative log-likelihoods. (B) Receptive field mosaics from one retina for the four major RGC types (ON parasol, OFF parasol, ON midget, OFF midget) used in the image reconstructions. Image quality metrics were computed over the shaded blue region, to exclude areas of the image that were insufficiently covered by receptive fields of recorded RGCs.

$$\hat{\mathbf{x}}(\mathbf{s}) = \arg\min_{\mathbf{x}} \big\{ -\log p(\mathbf{s} \mid \mathbf{x}) - \log p(\mathbf{x}) \big\}. \qquad (1)$$

The first term is the negative log likelihood of the stimulus, as obtained from an encoding model describing the probabilistic spiking of RGCs given a stimulus. In principle, any model can be used, and its parameters learned from experimentally measured RGC responses. The second term is the negative log prior of the stimulus which can be learned from natural images, independent of retinal responses. Because encoding models with varying levels of fidelity and detail can be mixed with priors of varying sophistication, the MAP approach allows us to probe the distinct roles of these two components in image reconstruction [24, 25].

### 3.1  RGC encoding models

Encoding models for each RGC must be fitted to the experimental data before performing MAP reconstruction. Two types of encoding models were used: (1) a linear-nonlinear-Poisson (LNP) cascade model with an exponential nonlinearity, the most commonly used model of RGC responses to visual stimuli [7]; and (2) a generalized linear model (GLM) that augments the LNP model with a feedback loop and cross-connections between neighboring cells, and which can accurately capture fine spike timing structure and cell-to-cell correlations [10]. Encoding models were fitted to a training partition of the data by minimizing the negative log-likelihood, and regularization hyperparameters were tuned by evaluating the negative log-likelihood on a test partition for a small subset of RGCs of each type.

**Linear-nonlinear-Poisson (LNP) encoding model**   The LNP model is the *de facto* standard model for describing the probabilistic spiking of RGCs in response to visual stimuli [7]. The model parameters for each RGC consist of a linear spatial filter $\mathbf{m}$, and a scalar bias $b$. In the model, a scalar generator signal $\mathbf{m}^T\mathbf{x} + b$ is passed through an exponential nonlinearity to yield a spike rate $\lambda(\mathbf{x})$. Responses, $s$, are defined as the RGC spike count in a 150 ms interval, which are assumed to be Poisson-distributed with rate $\lambda(\mathbf{x})$. The resulting negative log-likelihood for each RGC is

$$-\log p(s \mid \mathbf{x}) = \exp\{\mathbf{m}^T\mathbf{x} + b\} - s(\mathbf{m}^T\mathbf{x} + b), \qquad (2)$$

which is convex in $\mathbf{m}$ and $b$. In practice, to ensure that the spatial filters were spatially compact and corresponded approximately with receptive fields obtained using reverse correlation with white noise, the negative log-likelihood objective was augmented with two regularization terms: an L1 penalty to induce sparsity in the filter weights, and an L2 penalty to minimize the difference between the filter and the receptive field. The complete LNP-fitting objective function with both regularization terms is described in A.4. The parameters of each cell were optimized separately using FISTA [26].

MAP reconstruction requires the joint negative log-likelihood of the image based on the observed spikes from every RGC. Since the LNP model assumes that the responses for each cell are statistically independent, this is simply the sum of the single-cell negative log-likelihoods, which is convex in the image $\mathbf{x}$:

$$-\log p(\mathbf{s} \mid \mathbf{x}) = \sum_{i \in \text{Cells}} \exp\{\mathbf{m_i}^T \mathbf{x} + b_i\} - (\mathbf{m_i}^T \mathbf{x} + b_i). \tag{3}$$

**Generalized linear encoding models**   The generalized linear model (GLM) is an augmentation of the LNP model that incorporates the effects of spiking history and cell-to-cell correlations on neural response [10]. In the GLM, the $i$th RGC is parameterized by a spatio-temporal stimulus filter $k_i[x, y, t]$, a feedback (spike history) filter $f_i[t]$, a set of neighboring cell coupling filters $c_i^{(j)}[t]$, and a bias $b_i$. The GLM was fitted to spike counts measured within 1 ms time bins, a duration approximately matched to the spike refractory period [27, 28], yielding binary spike counts. To limit the number of parameters and improve computational efficiency, the stimulus filter was assumed to be space-time separable $k_i[x, y, t] = \mathbf{m}_i[x, y]h_i[t]$. Letting $*$ denote time-domain convolution, $w[t]$ denote the time course of the stimulus (boxcar), and $\{i\}^C$ denote the set of coupled neighboring cells, the generator signal $g[t]$ is

$$g_i[t] = (\mathbf{m}_i^T \mathbf{x})(h_i * w)[t-1] + (s_i * f_i)[t-1] + \sum_{j \in \{i\}^C} (s_j * c_i^{(j)})[t-1] + b_i. \tag{4}$$

Assuming a sigmoidal nonlinearity $\sigma(x) = e^x/(1 + e^x)$ and a Bernoulli spiking distribution, the negative log-likelihood used to fit a single cell (see A.5.2 for complete derivation) is

$$-\log p(s_i[N, ..., T] \mid \mathbf{x}, s_i[0, ..., N-1], \mathbf{s}_{\{i\}^C}[0, ..., T]) = \sum_{t=N}^{T} \left\{ \log(1 + \exp\{g_i[t]\}) - s_i[t]g_i[t] \right\}. \tag{5}$$

To simplify the GLM, the filters $h_i$, $f_i$, and $c_i^{(j)}$ were parameterized as weighted sums over a set of cosine-shaped "bump" functions [10] (discussed further in A.5.4). As with the LNP model, L1 and L2 regularization terms were added to constrain the spatial filters, and an additional $L_{1,2}$ group sparsity penalty was imposed on the coupling filters to eliminate spurious cell-to-cell correlations. The complete objective function is described in detail in A.5.3. Model parameters were found by alternating between convex minimization steps for the spatial and temporal filters of each RGC, using FISTA [26, 29] for each step.

For MAP image reconstruction, we compute the joint negative log-likelihood of the image based on responses of all cells over time bins $N, N+1, \ldots, T$. This can be constructed from the single cell negative log-likelihoods using the chain rule (see A.5.5 for derivation), and is again convex in the image $\mathbf{x}$:

$$-\log p(\mathbf{s}[N, ..., T] \mid \mathbf{x}, \mathbf{s}[0, ..., N-1]) = -\sum_{t=N}^{T} \sum_{i \in \text{cells}} \log p(s_i[t] \mid \mathbf{x}, s_i[0, ..., t-1], \mathbf{s}_{\{i\}^C}[0, ..., t-1]). \tag{6}$$

## 3.2   MAP with Gaussian $1/f$ priors

A stationary Gaussian prior with $1/f$ spectral energy is the simplest and most commonly used image prior [30], and is the basis for most classical image processing algorithms. The $1/f$ prior

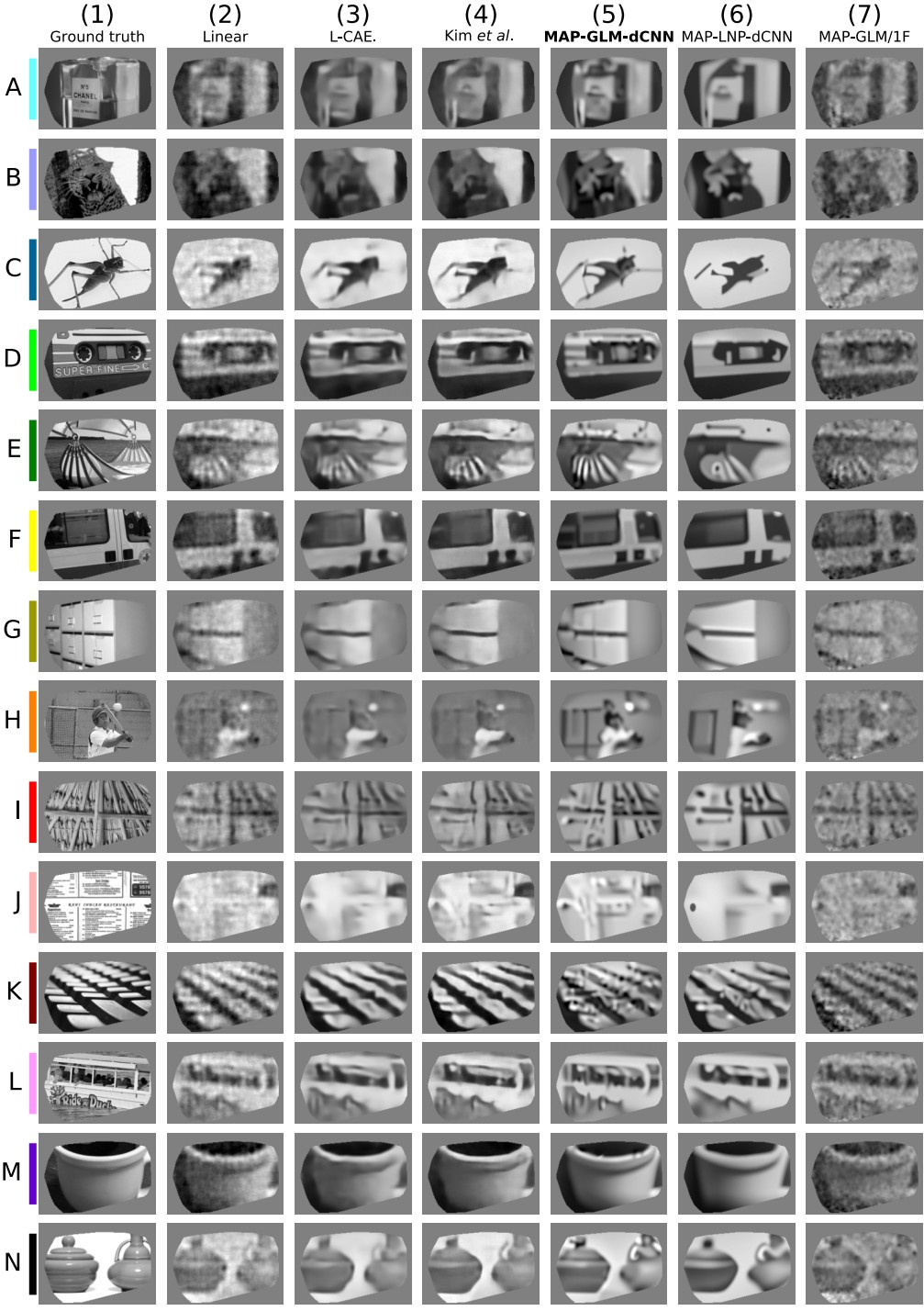

Figure 2: Selected stimulus images and reconstructions for the retina of Figure 1B. Images are masked to only include regions covered by recorded RGCs. Colors on the left correspond to the colored markers in the scatter plots in Figure 3. Column contents: (1) Ground truth stimulus image; (2) Linear reconstruction; (3) Linear reconstruction followed by CNN autoencoder (L-CAE, [8]); (4) Linear reconstruction followed by neural network regression [4]; (5) Our method, approximate MAP with GLM and denoiser prior (MAP-GLM-dCNN); (6) Approximate MAP with LNP encoder and denoiser prior (MAP-LNP-dCNN); (7) Exact MAP with GLM encoder and $1/f$ prior (MAP-GLM-1F).

assumes that pixels of the image are drawn from a stationary jointly Gaussian distribution (and thus that the spatial covariance matrix is diagonalized by the 2D Fourier basis) and that spectral power (variance of each spatial frequency component) falls off in inverse proportion to the square of the frequency $f^2$. Discarding terms that do not depend on the image $\mathbf{x}$, the negative log prior can be written as $-\log p(\mathbf{x}) = \sum_k |a_k(x)|^2/f_k^2$ where $a_k(x)$ is the amplitude of the $k^{\text{th}}$ Fourier coefficient at frequency $f_k$. MAP image reconstruction using this prior can be performed using standard unconstrained convex minimization methods because both the negative log prior and the RGC encoding negative log-likelihoods described in (3) and (6) are convex.

### 3.3 Approximate MAP with denoising convolutional neural network (dCNN) priors

Modern denoising convolutional neural networks can represent powerful image priors, but these priors are implicit [12, 31]: they are not expressed in closed form, and their values cannot be computed directly, making exact MAP inference difficult. The "plug-and-play" methodology provides an approximate iterative procedure for using such denoisers in MAP estimation problems [11], by incorporating them into variable-splitting optimization methods such as half-quadratic splitting (HQS) [32] or alternating direction method of multipliers (ADMM) [33, 34, 35]. Here, we use a method based on the HQS method presented in [15] to perform MAP reconstruction from RGC spikes. Following [15], we introduce an auxiliary variable $\mathbf{z}$, split the original problem in equation (1) into two complementary sub-problems, incorporate a regularization parameter $\lambda_p$ to control the strength of the prior term, and iteratively alternate between solving the two sub-problems:

$$\mathbf{x}^{(k+1)} = \arg\min_{\mathbf{x}} \left\{ -\log p(\mathbf{s} \mid \mathbf{x}) + \frac{\rho^{(k)}}{2} |\mathbf{x} - \mathbf{z}^{(k)}|_2^2 \right\} \tag{7}$$

$$\mathbf{z}^{(k+1)} = \arg\min_{\mathbf{z}} \left\{ -\lambda_p \log p(\mathbf{z}) + \frac{\rho^{(k)}}{2} |\mathbf{z} - \mathbf{x}^{(k+1)}|_2^2 \right\}. \tag{8}$$

Since the sub-problem in equation (8) has the same objective function as MAP Gaussian denoising with known noise variance $\lambda_p/\rho^{(k)}$, we compute its solution using a single forward pass of a pre-trained DRUNet denoiser network [15]. This is an approximation, because the denoiser was trained with an MSE objective and hence targets the mean of the posterior rather than its maximum. The approximation is correct in the limit of small noise, and thus can be used within an iterative procedure that gradually decreases the objective function [11]. The overall iterative MAP reconstruction procedure is given in Algorithm 1.

---

**Algorithm 1** Approximate MAP reconstruction from RGC spikes (MAP-*encoder*-dCNN)

1: Hyperparameters: $\lambda_p$, schedule $\rho^{(1)}, \rho^{(2)}, ..., \rho^{(N)}$
2: Inputs: observed spike count vector $\mathbf{s} \in \mathbb{R}^C$
3: Initialize $\mathbf{z}^{(1)}$ to linear solution
4: **for** $k \in 1, 2, ...K$ **do**
5:     $\mathbf{x}^{(k+1)} \leftarrow \arg\min_{\mathbf{x}} \left\{ -\log p(\mathbf{s} \mid \mathbf{x}) + \frac{\rho^{(k)}}{2}|\mathbf{x} - \mathbf{z}^{(k)}|_2^2 \right\}$
6:     $\mathbf{z}^{(k+1)} \leftarrow$ Unblind-DRUNet-Denoiser($\mathbf{x}^{(k+1)}; \sigma^2 = \frac{\lambda_p}{\rho^{(k)}}$)
7: **end for**

---

Unlike most applications of HQS, the encoding term in equation (7) is non-quadratic in $\mathbf{x}$ and hence (7) was solved iteratively (gradient descent with momentum and backtracking line search) rather than in closed form. $\mathbf{z}^{(1)}$ was initialized as the linear solution; using random Gaussian initialization does not significantly affect the results (see Appendix A.6). Because convergence in the mathematical sense is not necessary for most imaging applications [36], $K = 25$ iterations were used in Algorithm 1. As in [15], $\rho^{(k)}$ was increased per-iteration on a log-spaced schedule. The hyperparameters $\lambda_p$ and $[\rho^{(1)}, \rho^{(25)}]$ were determined by performing a grid search and evaluating reconstruction quality on an 80-image subset of the test partition. Variations in hyperparameters over a reasonable range ($\rho^{(1)} \in [10^{-2}, 10^{-1}]$, $\rho^{(25)} \in [30, 500]$, $\lambda_p \approx 0.1$) produced similar reconstruction quality, and optimal hyperparameters were similar across the two retinas and across LNP and GLM encoding models. Approximate MAP reconstructions using this algorithm are termed MAP-GLM-dCNN and MAP-LNP-dCNN for the GLM and LNP encoding models, respectively.

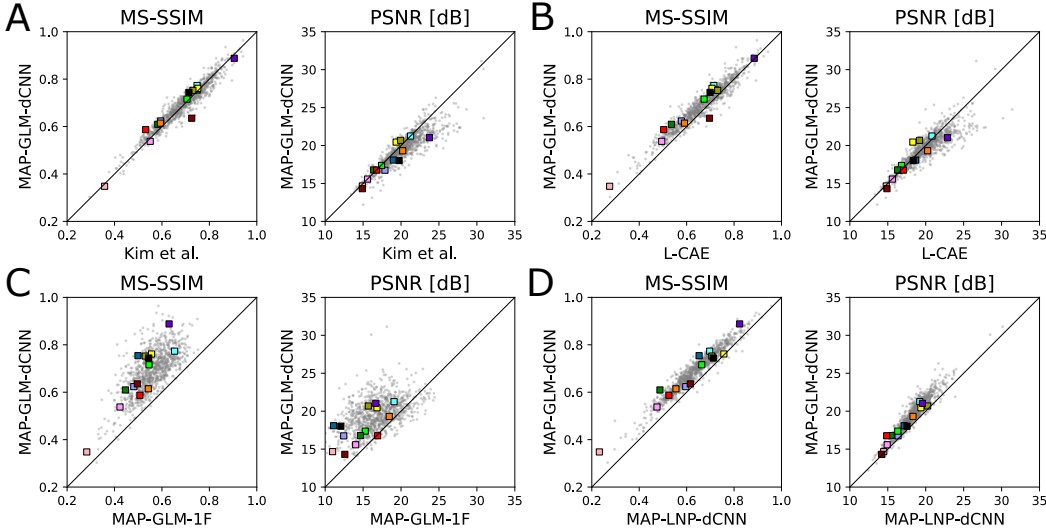

Figure 3: Comparisons of reconstruction quality of test data from spikes of the retina shown in Figure 1B. Each panel shows values of two metrics – multi-scale structural similarity (MS-SSIM) [37], and PSNR – for all test images (individual points). Colored markers correspond to the example images in Figure 2. Vertical axis of all graphs corresponds to our method (MAP-GLM-dCNN), and horizontal axis corresponds to other methods: (A) MAP-GLM-dCNN vs. Kim *et al.* linear with neural network regression [4]. MAP-GLM-dCNN achieves comparable perceptual similarity but worse PSNR (B) MAP-GLM-dCNN vs. L-CAE [8]. MAP-GLM-dCNN achieves improved perceptual similarity with slightly worse PSNR. (C) MAP-GLM-dCNN vs. MAP-GLM-1F. Reconstructions using the weaker $1/f$ prior have worse perceptual similarity and PSNR. (D) MAP-GLM-dCNN vs. MAP-LNP-dCNN. Reconstructions using the simpler LNP encoding model have worse perceptual similarity and PSNR.

### 3.4 Benchmark: nonlinear regression with artificial neural networks

Current state-of-the-art methods for reconstruction of natural images from RGC spikes rely on an initial linear reconstruction step [1, 3], followed by application of nonlinear neural networks to further improve the image. In one approach, Parthasarathy *et al.* [8] use a deep convolutional autoencoder (L-CAE) trained with MSE loss to incorporate image priors in the reconstruction. The authors trained and tested the model on simulated RGC spikes. For our comparisons, we used the published architecture and hyperparameters, but trained the model using backpropagation on experimentally measured retinal spike counts. Further details and characterization of the L-CAE on experimental data are provided in Appendix A.7. Another method was developed by Kim *et al.* [4], who partition the target image into high and low spatial frequency components. Linear regression from RGC spike counts is used to recover the low-frequency component, and a fully-connected neural network is used to non-linearly reconstruct the high-frequency component. The two components are summed and then passed through a final deblurring CNN. Both the high-frequency network and the deblurring CNN are trained with backpropagation on RGC responses to natural images. This method has achieved the most accurate reconstructions in the literature to date. Details of our implementation of the Kim *et al.* method are provided in Appendix A.8.

## 4 Results

### 4.1 Approximate MAP with GLM/dCNN matches or exceeds state-of-the-art results

To compare our MAP-GLM-dCNN method to current state-of-the-art approaches, image reconstructions were generated from the test partitions of the datasets. Example reconstructions are shown in columns 3-5 of Figure 2 for the L-CAE [8], for Kim *et al.* [4], and for our method, respectively. Qualitatively, the MAP-GLM-dCNN reconstructions are seen to be sharper than those of L-CAE, and contain additional image details (especially extended contours, as in rows C, E, G, H, I, L). When

Table 1: Average test and heldout MS-SSIM for each reconstruction method and retina.

| | L-CAE | | Kim *et al.* | | **MAP-GLM-dCNN** | | MAP-LNP-dCNN | | MAP-GLM-1F | |
|---|---|---|---|---|---|---|---|---|---|---|
| | test | held | test | held | test | held | test | held | test | held |
| Retina 1 | 0.665 | 0.661 | 0.687 | 0.681 | **0.689** | **0.688** | 0.635 | 0.636 | 0.557 | 0.552 |
| Retina 2 | 0.643 | 0.645 | **0.675** | **0.675** | 0.668 | 0.673 | 0.601 | 0.609 | 0.584 | 0.577 |

Table 2: Average test and heldout PSNR for each reconstruction method and retina.

| | L-CAE | | Kim *et al.* | | **MAP-GLM-dCNN** | | MAP-LNP-dCNN | | MAP-GLM-1F | |
|---|---|---|---|---|---|---|---|---|---|---|
| | test | held | test | held | test | held | test | held | test | held |
| Retina 1 | 19.9 | 20.1 | **20.2** | **20.5** | 19.5 | 19.5 | 18.3 | 18.4 | 16.8 | 16.8 |
| Retina 2 | 19.3 | 19.6 | **19.8** | **20.1** | 18.5 | 18.5 | 17.0 | 17.1 | 17.3 | 17.2 |

compared to Kim *et al.*, MAP-GLM-dCNN tended to recover more content, particularly straight edges (rows E, G, H, I, L), but sometimes exaggerated contrast (rows H, I, N). MAP-GLM-dCNN produced qualitatively different artifacts than the other methods. In particular, it sometimes hallucinated naturalistic structure not present in the stimulus images (rows J, K, N), including striking irregularities in contours (rows D, K, L, M).

Reconstruction quality for each method was quantified over the test and heldout partitions using MS-SSIM and PSNR. Scatter plots comparing MS-SSIM and PSNR on the test partition of one retina are shown in Figure 3A and 3B for Kim *et al.* and the L-CAE, respectively, and summary statistics over the test and held out partitions for both retinas are presented in Tables 1 and 2. On a per-image basis, the perceptual similarity (measured with MS-SSIM) of MAP-GLM-dCNN reconstructions to ground truth is higher than L-CAE (3B). The perceptual similarity of MAP-GLM-dCNN results is comparable to that of the much more complicated Kim *et al.* method (3A). But note that the PSNR of MAP-GLM-dCNN reconstructions was systematically lower than either benchmark. This is not surprising, as the MAP optimization procedure does not necessarily minimize MSE. These results held for both retinas (Tables 1 and 2).

### 4.2 Deep denoiser prior substantially improves image quality over $1/f$ prior

To test the importance of the image prior, MAP-GLM-dCNN results were compared against reconstruction using the GLM encoding model with the classical $1/f$ Gaussian prior (MAP-GLM-1F). Example reconstructed images using the denoiser prior and $1/f$ prior are shown in columns 5 and 7, respectively, of Figure 2. Images reconstructed with the denoiser prior are less "grainy", and tend to have both crisper edges and smoother surfaces. The artifacts seen in the $1/f$ examples are expected, since this simple prior does not constrain phase, whose alignment is essential for generating sharp spatially-localized features. Scatter plots of image quality on the test partition using MS-SSIM and PSNR are shown for one retina in Figure 3C, and mean values for both retinas are summarized in Tables 1 and 2. Consistent with the visual appearance, PSNR and MS-SSIM were systematically higher when using the denoiser prior, in both retinas. Thus, using the more sophisticated denoiser image prior substantially increased the perceptual similarity of the reconstructions to ground truth.

### 4.3 GLM encoding model recovers additional image structure over LNP encoding model

To test the importance of the encoding model, we compared images reconstructed using the GLM and LNP encoding models, both in combination with the the denoiser prior. Example images are shown in columns 5 and 6 of Figure 2. Images reconstructed using both models exhibit natural image structure, including smooth surfaces and well-defined edges, but the GLM-reconstructed images tended to have more realistic-looking textures, whereas the LNP-reconstructed images tended to be overly simplified. Moreover, the GLM method recovered more fine details (e.g., the legs of the insect in row C, the horizontal stripes on the tape cassette in row D, and the details on the hammock in row E, and the structure on the file cabinets in row G). The quality of image reconstructions for each image-response pair in the test partition for one retina were compared using MS-SSIM and PSNR in Figure 3D, and their mean values over the test and held out partitions for both retinas are summarized in Tables 1 and 2. In both retinas, images reconstructed using the GLM encoding model had systematically higher

MS-SSIM scores, indicating greater perceptual similarity to ground truth, than those reconstructed using the LNP encoding model. This demonstrates that the choice of encoding model significantly affects reconstruction quality, and that the inclusion of temporal spike dependencies and cell-to-cell correlations in the more sophisticated GLM encoding model provides important constraints on the visual signal encoded by the RGC spikes. This finding is consistent with previous work showing that decoding using the GLM (without priors) can access more information than simplified models lacking the cell-to-cell correlations or spiking history [10].

Table 3: Number of parameters trained on retinal data for each method (for retina 1). MAP-GLM-dCNN and L-CAE have comparable numbers of parameters. Because of sparsity regularization in the GLM spatial encoding filters, more than half (55%) of the parameters in the GLM model are zero. The Kim *et al.* model contains nearly an order-of-magnitude more parameters than either MAP-GLM-dCNN or L-CAE.

|  | L-CAE | Kim *et al.* | **MAP-GLM-dCNN** | MAP-LNP-dCNN | MAP-GLM-1F |
| --- | --- | --- | --- | --- | --- |
| Trained params. | $3.07 \cdot 10^7$ | $2.44 \cdot 10^8$ | $2.88 \cdot 10^7$ | $2.85 \cdot 10^7$ | $2.88 \cdot 10^7$ |

## 5  Discussion

We have presented a novel approximate MAP method for reconstructing natural images from the simultaneously recorded spikes of several hundred retinal ganglion cells, using an accurate probabilistic model of retinal encoding and a natural image prior implicit in a pre-trained denoising neural network. The method matches or outperforms the current state-of-the-art in terms of recovering naturalistic image structure and/or the perceptual similarity of reconstructions to ground truth, while also being more principled and interpretable due to the explicit Bayesian separation of the encoding model and prior. The new approach uses substantially fewer parameters than previous state-of-the-art methods based on neural networks, and does not require training neural networks on retinal data (the prior is obtained from a network trained exclusively on image denoising). We showed that both encoding model and image prior contributed to the high-quality image reconstructions: simplification of either substantially degraded performance. Thus, we expect that cell-to-cell correlations and temporal structure of spike trains, as well as sophisticated image priors, will prove important in understanding how the retinal signal is used by the brain.

Several previous studies have used GLM encoding models for stimulus reconstruction from experimentally-recorded retinal signals, revealing the significance of cell-to-cell correlations for decoding temporal structure in white noise stimuli [10, 38], and the significance of the temporal structure of spike trains in tracking moving features [2]. By including a complex natural image prior into a Bayesian reconstruction method, the present work more efficiently leverages both the GLM model and the experimental data to produce state-of-the-art natural image reconstructions.

The enhanced reconstructions and interpretability obtained with our method could lead to improved function of retinal implants for restoring vision. Previous work [5] has suggested that electrical stimulation with a retinal implant can be guided by minimizing the expected MSE of linearly reconstructed images. This method ignored fine temporal structure in RGC spike trains, and assumed that image priors captured by linear regression are sufficient for high performance. The method presented here offers an alternative approach to choosing simulation patterns to produce higher-fidelity artificial vision, while potentially being more robust than neural network methods. However, achieving this in real time with minimal latency presents a substantial technical challenge.

Our Bayesian reconstruction methodology relied on the convexity of the GLM encoding model, which allows for robust convergence. It is worthwhile to consider generalizations that might be used to evaluate more complex encoding models that are non-convex in the stimulus, such as subunit models [39, 40, 41] or artificial neural network retinal models[42, 43]. We are currently working on methods based on score-based sampling methods [44, 14] to answer this question.

Though the present work is limited to reconstruction of flashed static natural images from RGC spikes, extensions of the method could be used to probe how neurons encode visual information under more natural conditions. For example, a central problem is understanding how the visual system achieves high-acuity perception in the presence of "jitter" in eye position, even during fixation [45]. Previous computational efforts have probed this question, but have been largely limited to simulated data with

simple encoding models and stimuli [46, 47, 48]. Combining the methods put forth here with modern algorithms for image deblurring and motion-correction [49, 15] could yield more powerful methods to decode images from jittered retinal inputs. A related problem is understanding how the retina encodes the information contained in complex naturalistic movies [2], including movement of objects within a scene and other non-rigid transformations over time. The dimensionality of such stimuli and the consequent data requirements are high, so the use of machine learning methods to capture spatio-temporal priors [50, 51] separately from the retinal data will be important for understanding reconstruction in these contexts.

## 6 Funding

National Defense Science and Engineering Graduate (NDSEG) Fellowship (EGW), NSF IGERT 0801700 (NB), Wu Tsai Neurosciences Institute Big Ideas award, NSF CRCNS grant 1430348 (EJC/EPS), and NIH grants EY017992 and EY029247 (EJC).

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
