# OpenReview forum: "Maximum a posteriori natural scene reconstruction from retinal ganglion cells with deep denoiser priors"
_NeurIPS.cc/2022/Conference — NeurIPS 2022 Accept_

### Official Review · Reviewer_wKPR · 2022-07-06

**Rating:** 5
**Confidence:** 3
**Soundness:** 3 good
**Presentation:** 3 good
**Contribution:** 3 good

**Summary:**

The paper proposes a novel method for reconstructing natural images from recorded spikes of a several hundred retinal ganglion cells (RGC) by combining a generalized linear model (an augmentation of a linear-nonlinear-Poisson model)  encoded with the spike responses of the RGCs, along with image priors embedded in a deep, denoising CNN. Their combined model outperforms current state of the art that applies a deep convolutional autoencoder to linearly-reconstructed images.

**Questions:**

Why was the DRUNet denoiser network used specifically?

**Ethics Review Area:**

["I don’t know"]

**Limitations:**

The biggest limitation that I see is, as the paper states, achieving these sorts of results in the face of the “jitter” that is present in natural eye movements, and how to handle not only the jitter but also these sorts of complex image reconstructions in real time as an enhancement to biological vision, but these are all concerns that the authors do address, and satisfactorily considering the limitations of really just present computational power, especially in terms of enhancing biological vision presumably with a device capable of being worn.

**Strengths And Weaknesses:**

The primary strength appears to be the separation of concerns with regards to embedding of the image priors and the encoding of retinal spiking responses. Because the denoising convolutional neural networks (dCNN) used for the natural image priors are trained independently of the retinal spike training, the model becomes modular, thereby allowing a finer granularity for tuning parameters of either end of the model pipeline, or for testing wholly different models entirely. The results demonstrate clear progress on SOTA, but the larger contribution may be in improving the pipeline itself.

One weakness may be the use of minimizing the negative log likelihood of the posterior, ie max estimation likelihood(MLE), in the MAP reconstruction of the RGC spikes. Due to the mean-seeking behavior of MLE, it may be possible the model is learning to output false positives, ie attributing image prior reconstruction to RGC spikes when the spikes did not in fact contribute to that particular prior.

Another weakness may be the assumption of the application of the inverse square law with regards to the spike response, ie assuming a Gaussian distribution for the image priors. This combined with the fact that the images are not only static, but as the paper states, they do not account for natural eye movement. Possibly, natural vision inducing a response to RGC spikes does not follow a traditional Gaussian distribution (despite the mathematical simplicity it provides) due to the need for the spikes to account for such high variance in an incoming “image”, where an object or image may even be still, but the photons are being received into a highly dynamic, organic mechanism, namely the eyeball. And so the distribution of the photons may in fact have a fat tail distribution, rather than a normal one.

---

> ### Author Response · Authors · 2022-08-01
> **Response to Reviewer wKPR**
>
> **Regarding “assumption of the application of the inverse square law with regards to the spike response, ie assuming a Gaussian distribution for the image priors… ”:** Although the DRUNet denoiser network is *trained* to remove Gaussian noise [r2], the resulting learned natural image prior is highly non-Gaussian [r2, r3]. The use of Gaussian noise for training the denoiser allows for an interpretation of the relationship between the learned denoiser and the image prior, which has been exploited in the Plug-and-Play literature [r1], the Score-matching autoencoders, and the more recent “diffusion model” methods. Finally, the RGC encoding models in our paper assume Bernoulli or Poisson noise (Equations 3, 5, and 6), which are much more accurate representations of the biology than Gaussian noise. In summary, there is no Gaussian assumption for either the spiking response model or the image prior model used in our MAP formulation.
>
> [r1] Venkatakrishnan, S. V., Bouman, C. A. & Wohlberg, B. Plug-and-Play priors for model based reconstruction. in 2013 IEEE Global Conference on Signal and Information Processing 945–948 (IEEE, 2013).
>
> [r2] Zhang, K. et al. Plug-and-Play Image Restoration with Deep Denoiser Prior. IEEE Trans. Pattern Anal. Mach. Intell. 1–1 (2021) doi:10.1109/TPAMI.2021.3088914.
>
> [r3]  Z. Kadkhodaie and E. P. Simoncelli, “Stochastic Solutions for Linear Inverse Problems using the Prior Implicit in a Denoiser,” in Advances in Neural Information Processing Systems 34 (NeurIPS 2021), p. 13, 2021
>
> “**One weakness may be the use of minimizing the negative log likelihood of the posterior, ie max estimation likelihood(MLE), in the MAP reconstruction of the RGC spikes. Due to the mean-seeking behavior of MLE, it may be possible the model is learning to output false positives, ie attributing image prior reconstruction to RGC spikes when the spikes did not in fact contribute to that particular prior.”**
>
> The HQS method used in our method uses the concept of the Plug-and-Play literature, in which the solution to the MAP problem in equation (8) is approximated with an MMSE denoiser. This is typically justified by noting that it is correct in the limit of small noise, and thus can be used within an iterative procedure that gradually decreases the objective function [r1]. We will expand on this in the Discussion section.
>
> **Regarding eye movements:** We agree that handling eye movements is important, as mentioned in the Discussion section. Previous work has explored the joint estimation of stimulus and eye movements in *simulated* retinal ganglion cell populations [refs. 39, 40, 41 from the submission] using simplistic encoding models. To our knowledge there have been no studies of natural image reconstruction from *recorded* responses of retinal ganglion cells to jittered natural stimuli. We are interested in extending our framework to include joint estimation of stimulus and eye movements, but that analysis is far beyond the scope of the current paper.
>
> **“Why was the DRUNet denoiser network used specifically?”**  Any unblind denoiser with reasonable performance can be used with HQS algorithm presented in the paper. The DRUNet was used out of convenience (code availability); previous comparisons of DNN denoisers suggest performance is comparable amongst a number of different architectural choices [r3].

---

### Official Review · Reviewer_Etny · 2022-07-13

**Rating:** 8
**Confidence:** 5
**Soundness:** 4 excellent
**Presentation:** 4 excellent
**Contribution:** 3 good

**Summary:**

The authors propose an inference method for reconstructing images presented to retinas from the spiking activity of Retinal Ganglion Cells (RGCs). This is achieved using a Maximum A Posteriori (MAP) estimation where the posterior distribution is obtained from the combination of a likelihood modeling key aspects of RGCs (individually and as a network) and a prior for natural images.

The authors detail their models and optimization methods. Then they compare, both qualitatively and quantitatively, variation of their algorithms and also to previous state-of-the-art (SOTA) models. Their best model reaches SOTA performances on some scores.


**Questions:**

I am a bit surprised by the first sentence of the introduction. How do the authors know that only a small portion of visual information arriving at each eyes is transmitted to the brain ? I would have appreciated a reference here as "information" is a clearly defined concept in signal processing. Maybe the authors wanted to strike the reader by the existing contrast between the torrent of light or photons arriving at the retina and the somehow discrete nature of spiking activity ? Please add a ref or rephrase appropriately. The same remark hold for the first sentence of the abstract.

Apart from this minor remark, I think this is a well-written paper and a nice piece of work despite the results not being surprising.

**Limitations:**

The authors mention the negative impact of animal experimentation.
It is not clear if the data are openly accessible. If not, the authors should make them openly accessible to maximize the value of the data in many aspects (science, animals involved, ...)
The proposed work involve neural network training. The authors have used a single GPU but how long the training lasted for each model ? Does the performance gain justify the probable increase in energy consumption compared to other models ?

**Strengths And Weaknesses:**

Strengths :
- reconstructing images from neural activity is key to understand both natural images and neural activity, it is also a good task to evaluate models of biological neural networks,
- the methods used are recent or fairly recent and originally combined (FISTA, HQS, dCNN priors, ...),
- the results are good (reaching SOTA),
- the model could be used in combination with retinal implants to restore vision.

Weaknesses :
- low risk (dCNN priors have shown to be powerful in a variety of tasks)

---

> ### Author Response · Authors · 2022-08-01
> **Response to Reviewer Etny**
>
> We thank the reviewer for their comments.
>
> **Opening sentences in the abstract and introduction:** We intended these to be casual (and non-controversial) opening points about dimensionality. But we agree that there are important additional considerations (eg, graded vs. spiking responses), and our casual use of the term “information” is problematic in this context, where it might be interpreted as a more precise information-theoretic statement. We propose to soften as follows. Abstract: “Visual information arriving at the retina is transmitted to the brain by signals in the optic nerve, and the brain must rely solely on these signals to make inferences about the visual world.“ Introduction: “A torrent of visual information arrives at each of our eyes, but visual signals are represented by the spikes of a relatively small number of retinal ganglion cells (RGCs), suggesting that a large fraction of the image information is lost.“
>
> **“The authors have used a single GPU but how long the training lasted for each model?”** Fitting the encoding models for all ~700 cells in one dataset took about 24 hours with a single GPU. The training partition of the larger of the two datasets in the paper corresponds to approximately 2.5 hours of electrophysiological data per cell. Further improvements in the model fitting procedure (in particular, reducing the number of model parameters by cropping the spatial stimulus for each cell and using coarser time bins) has the potential to substantially reduce the fitting time without degrading the quality of the fits. This work is underway but will not be completed by the resubmission deadline.
>
> **“Does the performance gain justify the probable increase in energy consumption compared to other models?”** It is true that the computational cost of reconstruction of our method is greater than that of previously published methods, and the quantitative improvement as measured by MS-SSIM is small. We are working on improving efficiency.  But we would like to emphasize that the primary use of our method is not as an engineering solution for reconstructing images from RGC responses, but as a framework for scientific study of the properties of the retinal code, and their relationship to natural image priors.  In this context, computational efficiency is of lower priority than the quality of the encoding and prior models and the interpretability of the results.
>
> **“The authors mention the negative impact of animal experimentation. It is not clear if the data are openly accessible.”** We intend to make available the essential data required for reconstruction and the model parameters for doing so. However, we first intend to use the techniques developed here to probe several aspects of retinal signaling that motivated the studies (including how the timing of RGC spikes and correlations between cells influence the decoding) and publish this work in a journal article.

---

### Official Review · Reviewer_mbYL · 2022-07-14

**Rating:** 6
**Confidence:** 3
**Soundness:** 3 good
**Presentation:** 4 excellent
**Contribution:** 3 good

**Summary:**

Similarly to Parthasarathy/Batty 2017 and Warland 1997, this paper proposes a method to reconstruct visual stimuli from retinal ganglion cell electrophysiological measurements.

It proposes to combine a forward model and a prior in a MAP framework to obtain the stimulus image with maximum a posteriori probability.

For the forward model it proposes a baseline linear-nonlinear-Poisson model and the recurrent model from Pillow 2008. For the prior a frequency power law Gaussian prior and a denoising neural network prior (implicit through plug and play method) are used.
Combinations of forward model and and prior are evaluated.

It is shown that reconstructions using the recurrent model and the denoising neural network prior exceed other works in terms of SSIM similarity with the stimulus image.



**Questions:**

What about using the model in Batty, ICLR2017 "Multilayer recurrent network models of primate retinal ganglion cell responses" as a forward model?

In Parthasarathy/Batty 2017 it is shown for the convolutional autoencoder that its performance is strongly dependent on the data set used for training. My guess would be that this could be less the case for the denoising network used here. Could this be evaluated?

The abstract juxtaposes (non-)linear regression and MAP estimation, which are largely orthogonal concepts. Since even in this contribution a regression model is used for prediction, this should probably be rephrased.

**Limitations:**

Some limitations are discussed - I'd say this discussion is adequate. The scope of this work has no immediate negative societal impact, but may lead to some positive medical research outcomes down the line.

**Strengths And Weaknesses:**

This paper does solid work in combining existing methods into a new method that improves performance in the particular application of reconstructing stimuli from retinal ganglion cell activity.
In terms of numbers, the demonstrated improvement in terms of SSIM is marginal, but perceptually the reconstructions are more compelling than the alternative methods (my general experience with SSIM has been that there is no chance of it capturing such subtleties).

My main concern is about the interaction of injecting strong priors and the declared purpose of this method. If you are concerned about "how the brain might interpret the signal", is it a good idea to inject ever stronger naturalistic priors to fill in the missing information? Granted, the brain is good at such completion inference, too. But there is no reason to believe that the priors are in any way the same. Relatedly, I would be worried that the prior adds nonexistent details as briefly mentioned in the paper. While this does become visible in some of the reconstructions, it should also be noted that, reassuringly (or not?), it is not to an extent that the reconstruction metric against the ground truth is affected by it.

While the MAP framework is appealing because it allows designating a prior and a forward model, it also leaves an aftertaste of serving only the narration. E.g. it is a known pitfall to name L1-regularization a "sparsity prior", since only the MAP estimate will be sparse but no typical sample from the posterior distribution. All ideas presented here would also easily fall under the less semantically laden framework of regularized risk estimation, in which case the denoising step would replace the proximal operator of the regularizer.

---

> ### Author Response · Authors · 2022-08-01
> **Response to Reviewer mbYL**
>
> **“My main concern is about the interaction of injecting strong priors and the declared purpose of this method…”** The reviewer is correct that we must treat with caution the question of how the brain might interpret/process the signals from the retina, which is unknown and quite likely does not involve reconstruction of images. We also agree that we cannot know precisely what prior the brain has, or how strong it is. In the revision, we will ensure that the text is more clear and cautious about interpretations of what is happening in the brain.
>
> Our method does offer new ways to study retinal coding of natural images in a Bayesian framework. One could, for instance, hypothesize that the retinal code is well-matched to the distribution of natural images, in that the retina provides a lot of information about the stimulus when the stimulus is “natural”. We demonstrate that this may be the case by separating retinal coding from the image prior, and showing that reconstructed image quality was best when using both a good encoding model and a good prior, and that using less accurate models for either substantially degraded the reconstruction. Our approach also offers an intuitive way to evaluate in image domain how well various encoding models capture the retinal response to natural images, since we can evaluate reconstruction quality for each encoding model while holding the prior fixed.
>
> **“While the MAP framework is appealing because it allows designating a prior and a forward model, it also leaves an aftertaste of serving only the narration”.**  It is true that we rely on a “plug-and–play” style algorithm to iteratively achieve an estimate that we cannot guarantee is a unique MAP solution (since the proximal MAP step is replaced by a MMSE de​​noiser step, and the two will only be matched in the limit of small noise) - see r1, for example.  We will make this more explicit in the text, but still think it fair (and preferable) to stick with the “approximate MAP” terminology for the formulation and interpretation of the method. More generally, we appreciate the reviewer’s question regarding the juxtaposition of MAP and regression concepts - since our framework makes use of both, it is a bit tricky to explain.  By “regression” we mean methods that are trained directly (supervised) on the inverse problem (i.e., learn a mapping from spike trains to images, as in the Parthasarathy/Batty2017 paper), as opposed to a Bayesian formulation that combines a likelihood and prior whose parameters are fit separately to spiking responses or image databases, respectively (technically, an “empirical Bayesian” formulation). This is especially difficult to convey since our prior model is obtained from a denoiser that is trained using regression. We will adjust the text to unify and clarify the logic of these fitting procedures, and would welcome any further suggestions to make the distinction more clear!
>
> [r1] Venkatakrishnan, S. V., Bouman, C. A. & Wohlberg, B. Plug-and-Play priors for model based reconstruction. in 2013 IEEE Global Conference on Signal and Information Processing 945–948 (IEEE, 2013).
>
> **“What about using the model in Batty, ICLR2017?”:** Development of more sophisticated encoding models (ANN or otherwise) is an ongoing enterprise, and we expect that our paper provides a useful paradigm for evaluating these models.  That said, applying our HQS method to an ANN model (as found in the Batty paper) relies on solving equation (7), which may fail if the encoding negative log-likelihood is strongly non-convex. We are currently exploring generalization of sampling-based or diffusion approaches [r2, r3] as alternatives, and will mention this in the Discussion.
>
> [r2] Z. Kadkhodaie and E. P. Simoncelli, “Stochastic Solutions for Linear Inverse Problems using the Prior Implicit in a Denoiser,” in Advances in Neural Information Processing Systems 34 (NeurIPS 2021), p. 13, 2021
>
> [r3] Song, Y., Shen, L., Xing, L. & Ermon, S. Solving Inverse Problems in Medical Imaging with Score-Based Generative Models. Preprint at http://arxiv.org/abs/2111.08005 (2022).
>
>
> **“In Parthasarathy/Batty 2017 it is shown for the convolutional autoencoder that its performance is strongly dependent on the data set used for training…”:** The Parthasarathy/Batty 2017 results are based on *simulated* spikes and should be interpreted with caution, whereas our submitted paper uses spikes recorded from *ex vivo* retina. The question of how encoding varies across retinas is one that we hope to pursue, but is beyond scope of our current paper.  Note that previously published work on this topic [eg., r4] involves a massive increase in the number of data sets analyzed (hundreds rather than 2).
>
> [r4] Shah, N. P. et al. Individual variability of neural computations in the primate retina. Neuron 110, 698-708.e5 (2022).

---

> ### Comment · Reviewer_mbYL · 2022-08-09
> **Retaining score**
>
> Thanks for the detailed responses to all reviewers.
>
> I have decided to retain my score, as my overall assessment of this paper has not dramatically changed.

---

### Meta-Review · Area_Chair_xqEy · 2022-09-07

**Recommendation:** Accept
**Confidence:** Certain

**Metareview:**

The paper proposes a MAP method for natural scene reconstruction from population recordings of RGCs which combines a deep image denoiser with a simple generalized linear neural encoding model. The results are shown to be state of the art, and the method is seen as novel and a useful contribution to the community.

The reviewers unanimously recommend acceptance.

**Award:**

No

---

### Decision · Program_Chairs · 2022-09-14

Accept